# OpenReview forum: "UnlearnCanvas:  Stylized Image Dataset for Enhanced Machine Unlearning Evaluation in Diffusion Models"
_NeurIPS.cc/2024/Datasets_and_Benchmarks_Track — NeurIPS 2024 Track Datasets and Benchmarks Poster_

### Official Review · Reviewer_T5uq · 2024-07-22
**A Review of Its Contribution to Machine Unlearning in Diffusion Models**

**Rating:** 7
**Confidence:** 3
**Clarity:** The paper is well written.

**Review:**

This paper introduces the "UnlearnCanvas" dataset designed to assess the MU capabilities of DM in style and object unlearning. It offers a dataset with high-resolution images and detailed class labels, coupled with eleven new benchmarks to evaluate nine machine unlearning methods. This approach tries to advances the field's methodology.

**Pros:** (Ref: Strength)
1. **Dataset and Benchmarks:** Provides a dataset with detailed classifications that facilitates precise evaluations of unlearning.
2. **Application to Style Transfer:** Demonstrates versatility by extending to style transfer applications.
3. **Evaluation Depth:** Utilizes a comprehensive benchmark set for extensive evaluation.

**Cons:** (Ref: Limitations)
1. **Style Variability:** The consistency of style representation due to reliance on Fotor for image generation.
2. **Style Scope Limitation:** The dataset's focus on predefined styles may limit its broader applicability.
3. **Unclearness of benchmark results:** The metrics results(Fig.1 (b)) illustrates varied metric results, revealing the current judgment rules’ lack of clarity and distinctiveness, which complicates consistent application.
4. **Biased Classifier:** Benchmarks depend on how each model performs well in **classification**, and raises concerns about how the performance could be affected by ***how close the data's features are to the classifier's data***.
5. **Novelty:** It's difficult to discern the novelty compares to the previous work[[1](https://unlearning-challenge.github.io/assets/data/Machine_Unlearning_Metric.pdf)].

The paper provides advancements in evaluating MU for DM, recommending an expansion in style diversity and improved benchmark usage guidance for future research.

**Strengths:**

1. The Author suggests a number of benchmarks based on a rational analysis of performing MU on DM and has applied them to various MU methods.
2. Explicitly defined class labels (Sketch, Crayon, ...) enable ease of measuring how objects are retained well.
3. Contains high-resolution images compared to the conventional used dataset.
4. The author extends their method to the domain of style transfer, which also lacks consensus on performance measurement.

**Additional Feedback:**

I would be inclined to **raise my score** if my concerns resolved.

**Correctness:**

Experimental setup for benchmarks are logically sound, while contains some concerns. _(Please refer the limitation section)_

**Documentation:**

Dataset provided with the complete of the documentation

**Limitations:**

1. Can the images constructed via Fotor be considered the GT stylized image? I am curious whether the quality made by Fotor is good enough to represent the style. There can be various types within a single style. For instance, Van Gogh's style contains "The Starry Night", mainly dark and blue, and "Sunflowers", mainly yellow and light. Can images made by Fotor(=UnlearnCanvas) effectively implicate the variety of these styles?
2. (~Q1) While the total 60 classes of the style seem sufficient, I wonder if the author has considered it the case that they want to unlearn the style out of the set.
3. Figure 1(b) presents the results of various applied metrics, raising concerns about the current judgment rules’ lack of clarity. This deficiency makes it challenging to consistently apply these rules across different cases. For instance, in the Cross-Domain (CRA) section, the Salun model demonstrates strong performance, while in the IRA section, the SEOT model excels. However, in the MU domain, it is crucial to enhance both models’ performance. To make the benchmark truly useful, I hope the author provides more guidance, such as returning a final score by providing a weighted sum of each metric based on their importance.
4. The benchmarks depend on how each model performs well in classification, so they depend on the specific classifier. This raises concerns about how the performance could be affected by how close the data's features are to the classifier's data, which can be critical for ability of benchmarking.
5. Previous research[[1](https://unlearning-challenge.github.io/assets/data/Machine_Unlearning_Metric.pdf)] shows the metrics to show forgetting quality and retainability. I hope the authors can emphasize their work's contribution compares to it.

reference:
[1] [NeurIPS Unlearning Competition](https://unlearning-challenge.github.io/assets/data/Machine_Unlearning_Metric.pdf)

**Opportunities For Improvement:**

As the author mentioned, the performance of generation/classification of the model(ViT) trained with UnlearnCanvas exceeded one with WikiArt(Tab. A3,4). Here, does the case of detecting copyright infringement [ref: line134] also show the outperformance compared to WikiArt so that it can show its broader use cases not only for the MU?

**Relation To Prior Work:**

The dataset effectively illustrates how this work differs from previous contributions. However, it would be beneficial if the authors could expand their discussion on the benchmarks.[[1](https://unlearning-challenge.github.io/assets/data/Machine_Unlearning_Metric.pdf)]

**Summary And Contributions:**

This paper constructed a **dataset** named UnlearnCanvas, which mainly focused on style/object unlearning on the diffusion model. The Dataset consists of the seed images(= base "object" image, 20 images per class, total 20 classes) and stylized images(= 60 diverse stylized images per class, constructed via Fotor*).
The authors raised three challenges that communities faced in the Style/Object MU-Diffusion field:
1. The absence of a consensus on an unlearning target (such as A for style unlearning, B for objects unlearning)
2. Lack of study on retainability of the model after the unlearning process + Lacks of object labels
3. Low performance of DM-generated images and even lower after the style is applied

Then, the authors suggest UnlearnCanvas, with three corresponding advantages:

1. Prepare for all situations (Style-Object dual supervision)
2. Add in/cross-domain retainability analysis (IRA, CRA, respectively) + Suggest explicit label.
3. Accuracy enhancement via UnlearnCanvas

The author also suggests eleven **benchmarks** (UA, IRA, CRA, etc.) with corresponding results of nine MU-diffusion models. They expand their usage in Style-Transfer in three pipelines. (Style Authenticity, Content Integrity, Algorithm Stability)

---

> ### Author Rebuttal · Authors · 2024-08-16
>
> # Point-to-Point Response to Reviewer T5uq
>
> We thank the reviewer for the constructive suggestions. Below, we present a point-to-point response.
>
> **Q1. Can the images constructed via Fotor be considered the GT stylized image? Is the quality by Fotor good enough to represent the style? Can images via Fotor effectively implicate the variety of the styles (e.g., "The Starry Night" and "Sunflowers" in Van Gogh's style contains)?**
>
> **A1.** This is an insightful question regarding the variety within artistic styles. We want to make the following clarifications.
>
> First, although it is possible to include different types in one style, we intended to keep high intra-style consistency to ease the precise evaluation challenge of unlearned DMs (C3 in Sec. 2 of our submission). We hope to impose a clearly-defined and controllable knowledge in UnlearnCanvas.
>
> Second, as pointed out by the reviewer, Fotor is capable of generating images using different style references. Taking Van Gogh style as an example, **Fig. R1** ([link](https://imgur.com/a/O3t3nJz)) shows examples including "The Starry Night," "Cafe Terraces," "The Starry Night over the Rhone," and "Sunflowers." Thus, it is possible to include multiple types within a single style. On the other hand, "Sunflowers" shows higher stylistic discrepancy compared to the other three types. Therefore, UnlearnCanvas did not include all the types within a style but choose types selectively for precise post-unlearning evaluations. We will add this insightful discussion in the revision.
>
> **Q2. While 60 styles seem sufficient, can authors unlearn the style out of the set?**
>
> **A2.** Excellent question! We believe the current setup still supports the evaluation of styles out of UnlearnCanvas. Specifically, we have observed that the SD v1.5 model, even finetuned on UnlearnCanvas, retains the ability to generate styles outside UnlearnCanvas (e.g., Rembrandt styles). Therefore, external styles can theoretically be used as unlearning targets.
>
> Yet, the lack of accurate quantitative evaluation systems for such styles is still a significant challenge, as shown in Sec. 2. Training a reliable classifier for the Rembrandt style, for example, is difficult due to the lack of annotated data, which makes evaluating the post-unlearning performance precisely more challenging when dealing with styles outside the predefined set.
>
> **Q3. I hope the author provide more guidance like a final score for each method.**
>
> **A3.** Thanks for this suggestion. In our submission, we did not adopt a single-score evaluation because it may obscure important details revealed by our benchmark. With Figure 1(b)/Table 2, method designers and users can clearly identify the strengths and weaknesses of each method and make informed decisions accordingly.
>
> Meantime, we also admit a single-score evaluation is helpful for directly comparing two methods. Therefore, in line with your suggestion, we adopt a ranking-based approach to compare different methods using harmonic mean calculated for each method based on their ranking in all individual metrics. The results are shown in **Tab. R4** ([link](https://imgur.com/a/QUbqWB2)). From above, the best method is SalUn with a final score of 2.18, with ESD ranking the second place. We will include this discussion in our revision.
>
> **Q4. The benchmarks depend on the specific classifier. How could the performance be affected by how close the data's features are to the classifier's data?**
>
> **A4.** There might be a misunderstanding of the training of classification models in our work. To clarify, the style/object classifier is directly trained on UnlearnCanvas. Thus, the classifier’s training is independent of the diffusion models. Accordingly, all the diffusion models post-unlearning can be evaluated using the classifiers.
>
> **Q5. NeurIPS Unlearning Competition shows the metrics for forgetting quality and retainability. I hope the authors can emphasize their work's contribution compared to it.**
>
> **A5.** Thanks for this important literature. Our contributions differ from [1] in several key ways:
>
> 1. **Tasks**: [1] focuses on classification tasks, while our work centers on image generation. This difference leads to fundamentally different challenges, even though forgetting quality and retainability are tested in both work.
>
> 2. **Metrics**: In our work, we for the first-time introduced the cross-domain and in-domain retainability, metrics not previously explored. Additionally, we evaluate FID to assess the quality of generated images and include three efficiency metrics to comprehensively depict efficiency—none of which are covered in [1].
>
> 3. **Challenging tasks**: Our benchmark includes several challenging tasks, such as unlearning with different granularities, robustness, and sequential unlearning. These reveal important and previously unknown challenges, highlighting the limitations of existing methods and suggesting meaningful future directions with our dataset.
>
> **Q6. Can this dataset be used for copyright infringement detection (CID) like [2]?**
>
> **A6.** Thanks for pointing this out! We believe UnlearnCanvas can be used for CID but relies on our predefined style set.
>
> In [2], CID is achieved by precise classification on the artistic styles. In our work, the precise style or object-wise classification on UnlearnCanvas can be achieved easily by its design for an accurate unlearning evaluation. Thus, if we treat the styles defined in UnlearnCanvas as the copyrighted styles, it can be used for effective CID, just as [2] did on WikiArt. It can also be used to evaluate if the  infringement is alleviated after unlearning. However, we also admit, just like WikiArt, UnlearnCanvas still relies on the predefined style set, therefore can be difficult for cases where an open set of styles are used. We appreciate this insightful question and will discuss it in the revised "Conclusion, Discussion, and Limitation" section.
>
> > [2] Rethinking Artistic Copyright … Models

---

> > ### Comment · Reviewer_T5uq · 2024-08-30
> >
> > I'm satisfied about the responses from authors, so I raised my previous score.

---

> > > ### Author Response · Authors · 2024-08-30
> > > **Thank you and inquiry**
> > >
> > > Dear Reviewer T5uq,
> > >
> > > Thank you for your thoughtful feedback on our submission. We are pleased to know that you found our responses satisfactory.
> > > In the ‘Additional Feedback’ section of your review, you mentioned, “I would be inclined to **raise my score** if my concerns were resolved.” Based on this, we kindly inquire if there is a possibility for you to consider and possibly increase your score. If there are any remaining concerns or aspects that require further clarification, please feel free to let us know. We are prepared to address any additional comments you might have.
> > >
> > > Thank you once again for your time and careful consideration.
> > >
> > > Best regards,
> > >
> > > Authors

---

> ### Author Response · Authors · 2024-08-28
> **Thank you and genuine request for follow-up discussions!**
>
> Dear Reviewer T5uq,
>
> We extend our heartfelt appreciation for your dedicated review of our paper. As outlined in the recent NeurIPS email communication, we are fully aware of the commitment and time your review entails. Your efforts are deeply valued by us.
> During the rebuttal, we have made substantial effort in addressing all the reviewers’ questions, e.g., the evaluation using another diffusion models, the variety of the styles, adding new methods to the benchmark, the quantitative comparison between UnlearnCanvas and WikiArt; see a summary of extended experimental justifications/results in [general response](https://openreview.net/forum?id=t9aThFL1lE&noteId=IyUyjiXAJh).
>
> As the author-reviewer discussion phase draws to an end soon, we wish to extend a respectful request for your feedback about our general response and individual responses. Your insights are of immense importance to us, and we eagerly anticipate your updated evaluation. Should you find our responses informative and useful, we would be grateful for your acknowledgment. Furthermore, if you have any further inquiries or require additional clarifications, please don't hesitate to reach out. We are fully committed to providing additional responses during this crucial discussion phase.
>
> We sincerely thank you for your continued support and consideration. Your expertise is pivotal to the advancement of our research.
>
> Best regards,
>
> Authors

---

### Official Review · Reviewer_MenV · 2024-07-23
**UnlearnCanvas: Stylized Image Dataset for Enhanced Machine Unlearning Evaluation in Diffusion Models**

**Rating:** 5
**Confidence:** 4
**Clarity:** Yes.

**Review:**

The motivation is clear and this paper is well-written. According to the authors, it is the first benchmark dataset with precise evaluation criteria to support a comprehensive, quantitative, and precise assessment of unlearning performance.

**Strengths:**

The contribution is significant. This paper provides a detailed experimental setup and metrics for a thorough assessment of MU methods of DMs, which is relevant to the broader research community.

**Additional Feedback:**

No.

**Correctness:**

This paper proposes a benchmark which recollect and preprocess the datasets. 7 quantitative metrics is provided. My concern is that the evaluation pipeline is based on the fine-tuned DM and trained vision transformer as above stated.

**Documentation:**

The documentation is complete. I am curious about the choice of style and objects, and the number of each.

**Ethics:**

Data quality and representativeness.

**Limitations:**

The authors didn’t mention about the limitations and potential negative societal impact. In my point of view, the definition and choice of style and object is not clear and more like heuristic, which may results in and inability to adequately test the methods.

**Opportunities For Improvement:**

Though the authors highlight the intra-style coherence and inter-style distinctiveness, will it mean that the difficulty of unlearning is degraded? For example, in Figure A1, the “Van Gogh” style images have similar tone, which is different from the commonly recognized “Van Gogh” style, so as the “Picasso” style. Will this characteristic influence the comparison of unlearning methods?

The unlearning methods are applied on the fine-tuned specific DM, and the results are evaluated by the trained vision transformer. To demonstrate that the results are equivalent to the results on other models, perhaps further experiments could be conducted on a wider range of models to illustrate the generalizability.

**Relation To Prior Work:**

Yes.

**Summary And Contributions:**

This paper introduces the UNLEARNCANVAS dataset that for evaluating the unlearning of artistic styles and associated objects. This paper addresses the need for a standardized, automated quantitative evaluation framework for MU methods, identifying weakness in current evaluation practices.

---

> ### Author Rebuttal · Authors · 2024-08-16
>
> # Point-to-Point Response to Reviewer MenV
>
> We sincerely thank the reviewer for the constructive suggestions. Below, we provide point-by-point responses to each of your questions.
>
> **Q1. Will the high intra-style consistency degrade the unlearning difficulty? Will this characteristic influence the comparison of unlearning methods?**
>
> **A1.** This is an excellent question. We believe that higher intra-style consistency does not necessarily reduce the difficulty of the unlearning task and still ensures a fair comparison among different methods. There are two main reasons for this:
>
> * **Entangled knowledge in diffusion models**: While intra-style consistency simplifies the evaluation process, it does not necessarily make unlearning easier. In fact, the knowledge of these styles stored in diffusion models can still be strongly entangled. This entanglement can result in high difficulty in maintaining retainability. For example, in Figure 5 (left) in the submission (see [here](https://imgur.com/a/JDkoncJ)), certain styles, such as Crayon, can influence others when being unlearned, as indicated by a dark row in region B1. This demonstrates that, despite visual distinctiveness, challenges can arise when attempting to erase knowledge from diffusion models.
>
> * **A pressing need for precise evaluation**: As we highlighted in Sec. 2 of our submission (Challenge C3), a main challenge with current unlearning methods for diffusion models is the imprecise evaluation of concept erasure, particularly due to the absence of supervision for unlearning concepts like styles and objects and the lack of accurate prediction models for these concepts. To address this, we intend to implement intra-style consistency to enhance the reliability of unlearning evaluations.
>
> **Q2. Only one diffusion model and classification model is used in the benchmark. Authors should test a wider range of models to illustrate generalizability.**
>
> **A2.** We appreciate this constructive question. To alleviate your concern, we conducted additional experiments on Stable Diffusion (SD) 2.0 and SD v1.4, which are the newer and older versions of the SD 1.5 used in our original submission. Please refer to our general response (**GR** ([link](https://openreview.net/forum?id=t9aThFL1lE&noteId=IyUyjiXAJh))) for more details and the results are reported in **Table R1** ([link](https://imgur.com/a/qmOe1Dh)).
>
> The additional experimental justification demonstrates that the major conclusions presented in Table 2 remain consistent, regardless of the diffusion model type tested. The strengths and weaknesses of each unlearning method, as identified in Table 2, also persist across different SD models. Notably, the low IRA (In-Domain Retaining Accuracy) and CRA (Cross-Domain Retaining Accuracy) values in SD v2.0 and SD v1.4 implies the importance of retainability for  a comprehensive evaluation of the performance of the unlearned diffusion models, consistent with our discussion in Lines 263-267. In addition, we also observe that no single method excels across all metrics simultaneously, consistent with our discussion in Lines 279-283.
>
> In addition, we also trained different types of style- and object-classifier to evaluate the answer from the unlearned model, to demonstrate that our conclusions are not affected by the choice of classification model. Specifically, we trained a new ResNet-101 (differentiated from the ViT-Large used in the original submission) and re-evaluated the results in Table 2. The comparative results are shown in **Table R3** ([link](https://imgur.com/a/WIHRpoc)). As seen, the result difference incurred by the change of the classification model is negligibly small ($< 1%$). This indicates that the findings in Table 2 are robust to different classification models.
>
> We hope this additional experiment addresses your concerns about the generalizability of our conclusions across different SD model types.
>
> **Q3. The authors didn’t mention the limitations and potential negative societal impact.**
>
> **A3.** We apologize for any confusion. We discussed the potential societal impact in Appendix G, and at the time of submission, we did not identify any negative impacts. However, upon further reflection, we recognize that the unlearning insights revealed in this work could also be used by adversarial attacks against  the unlearned model. We appreciate the reviewer's suggestion and will include a discussion of these potential risks in our revisions.
>
> **Q4. In my point of view, the definition and choice of style and object is not clear and more like heuristic, which may result in inability to adequately test the methods.**
>
> **A4.** We apologize for any lack of clarity in our initial explanation regarding the selection of styles and objects for our dataset. We chose styles that maximize visual differences to ensure clear inter-style discrepancies and selected commonly seen objects like dogs, cats, and trees to maintain relevance to typical image datasets. This finite selection is also considered for supporting the computational feasibility in unlearning and evaluation of diffusion models. We believe our dataset, with its diverse and substantial number of style-object combinations (e.g., Table 3 in our submission), is well-suited to rigorously test unlearning methods and provide precise evaluations.

---

> ### Author Response · Authors · 2024-08-28
> **Thank you and genuine request for follow-up discussions!**
>
> Dear Reviewer MenV,
>
> We extend our heartfelt appreciation for your dedicated review of our paper. As outlined in the recent NeurIPS email communication, we are fully aware of the commitment and time your review entails. Your efforts are deeply valued by us.
> During the rebuttal, we have made substantial effort in addressing all the reviewers’ questions, e.g., the evaluation using another diffusion models, the variety of the styles, adding new methods to the benchmark, the quantitative comparison between UnlearnCanvas and WikiArt; see a summary of extended experimental justifications/results in [general response](https://openreview.net/forum?id=t9aThFL1lE&noteId=IyUyjiXAJh).
>
> As the author-reviewer discussion phase draws to an end soon, we wish to extend a respectful request for your feedback about our general response and individual responses. Your insights are of immense importance to us, and we eagerly anticipate your updated evaluation. Should you find our responses informative and useful, we would be grateful for your acknowledgment. Furthermore, if you have any further inquiries or require additional clarifications, please don't hesitate to reach out. We are fully committed to providing additional responses during this crucial discussion phase.
>
> We sincerely thank you for your continued support and consideration. Your expertise is pivotal to the advancement of our research.
>
> Best regards,
>
> Authors

---

### Official Review · Reviewer_CJtq · 2024-07-25

**Rating:** 9
**Confidence:** 5
**Correctness:** The conclusions in this paper are in …

**Review:**

Below is a summary of the evaluation of this paper from different perspectives.

* **[Quality]**: The quality of this paper is high in terms of paper writing, figure plotting, motivation illustration, experiment design, and analysis of the results in many novel perspectives.

* **[Clarity]**: The paper enjoys high-quality writing and high clarity, including the motation, the introduction of the dataset and the evaluation framework, as well as the rationale behind the experiment design. In the meantime, this paper contains several typos, which can be improved. The clarity of the paper can also further be improved by simplify some of the notations in this work, which may make them accessible to readers that are not the expert in the field of unlearning.

* **[Significance of this work]**: This paper is of high significance in terms of both the dataset and the benchmark. The dataset bridges the gap of the need for a higher quality dataset taylored for diffusion quantitative evaluation. The significance of the benchmark lies in its well-defined evaluation framework, as well as its novel insights from the unique analysis in this work. The benchmark is a milestone for the field of diffusion unlearning, and provides the readers with a holistic view on the pros and cons of different methods, as well as the challenges this community is facing.

In the next, as the main body of my review, I will provide a detailed discussion on the strengths and weaknesses of this work.

First, this paper contains the following strengths:

* **[A Clear Demonstration of the Existing Challenges.]** Discovering questions is usually more difficult and also more significant than solving them. This work first presents a summary of the development of the field of diffusion unlearning and concept erasing, and then points out several key challenges that this field is facing. I in general agree with the challenges proposed by the authors.

* **[The Introduction of a New Dataset.]** The new dataset is in the form of image pairs. This is great compared to many other datasets, as such supervised style provides the “ground truth” that are required and valued in many tasks. The balanced struction of the dataset comes

* **[A Novel Design of New Metrics for Diffusion Unlearning.]** The new metrics, namely the introduction of cross-domain and in-domain retainability, is for the first-time brought up to the table and I agree that the retainability should be considered carefully. This dissection on the retainability indeed can lead to more previously unknown but important findings on the pros and cons of different unlearning methods.

* **[The Introduction of the new tasks in the benchmark]**. The proposed benchmark includes three more challenging forgetting tasks, which are novel and the insights from them are representative and interesting.

* **[Extensive Experiments]**: This benchmark tested 9 SOTA methods, and conducted very detailed analysis on the strengths and weaknesses of each method.

* **[Nice Figures]**: The figures in this paper are nicely plotted, clear and informative. I particularly love Figure 5, which are very clear but at the same time super informative. Such figures can greatly help the readers catch the relationship between IRA and CRA.

* **[Novel Insights]** The insights revealed in this paper’s experiments are very novel, meaningful to the community and this field.

In the menatime, this work also suffers from the following weaknesses:

* **[The necessity of fine-tuning.]* Although this dataset is able to evaluate different unlearning methods, it requires the model prior to unlearning to be finetuned on the proposed dataset. This is not convenient.

* **[Missing Related Literature]** There is an important literature missing from discussion and comparison:

> MACE: Mass Concept Erasure in Diffusion Models, CVPR 2024

* **[Quantitative Judgement of Stylistic Evaluation]**: The authors claimed high stylistic consistency and high stylistic discrepency of the proposed dataset compared to existing ones, such as WikiArt. However, the authors did not prove it. Are there quantitative methods to validate this argument?

* **[Implementation Details of the Benchmarked Methods]**: Many methods benchmarked in this paper have many variations. For example, ESD has the ESD-x and ESD-u. I am wondering what versions of the different methods were tested in the benchmark. The authors should also make this clear in the paper.

* **[Question on Figure 6]**: Does the results shown in Figure 6 indicate that unlearning styles are more difficult than objects, because the unlearning accuracy drops in style unlearning seem to be more siginificant than that of object unlearning. However, this conclusion seems to be contradictory to some existing literature. For example, in ESD, unlearning objects seem to be simpler.  Why is this happening?

* **[Typos]** There seems to be a typo in Figure 1’s caption, and the names of different metrics are very confusing. First, wouldn’t “the retainability of innocent knowledge (IRA, CRA, FR, **FU**);” be “the retainability of innocent knowledge (IRA, CRA, FR, **CR**); ”? Second, why are the abbr. of the metrics for the sequantial unlearning dubbed “CR”and “CU” instead of “SR” and “SU”? Wouldn’t this make the names more consistent? Further, in order to achieve consistency, it would be better to mark the number “27.8” for EDiff (Memory Efficiency) in Table 2 in red. At Line 339, the abbreviation for “Object consistency” should be “OC” instead of “BC”.

* **[Possible Solutions or Directions of the Discovered Challenges]**: In the Experiment section, the authors demonstrated several challenges residing in the existing literature from different unlearning settings, such as the fine-scale unlearning and sequential unlearning. However, I think it would be better to discuss how these challenges could be solved and possible meaningful research directions in the future. This would be a plus to the community.

**Strengths:**

This paper has several strengths, which are summarized in the “Review” section.

**Additional Feedback:**

I do not have additional feedback.

**Clarity:**

This paper is carefully drafted, well organized, and clearly articulated. There are some typos, which were reported in the “Review” Section. On top of that, the paper follows the motivation-solution-insights-challenges and each part is well positioned and the article is coherently presented.

**Documentation:**

The proposed dataset is well documented. The generation and collection of the dataset is clearly disclosed. The maintainance plan is also clear. The dataset sheet is also provided.

**Ethics:**

N.A.

**Limitations:**

This paper clearly discussed its limitations in the last section. This paper also discussed its possible (positive) societal influences.

**Opportunities For Improvement:**

These are summarized in the “Review” section.

**Relation To Prior Work:**

In general, this work did a very good job surveying existing diffusion unlearning task and Table 1 made a very good summary on the metrics and scopes of the existing literature. There is nevertheless a literature missed, which I have pointed out in the “Review” sectino.

**Summary And Contributions:**

This paper addresses the challenges in evaluating machine unlearning for diffusion models, particularly in mitigating the generation of harmful content and addressing copyright issues. The major research question revolves around establishing a robust, standardized evaluation framework for DM unlearning. The paper's main contribution is the introduction of UNLEARNCANVAS, a high-resolution stylized image dataset that allows for the comprehensive evaluation of unlearning artistic styles and associated objects using seven quantitative metrics. Extensive experiments benchmark nine state-of-the-art MU methods, uncovering insights into their strengths, weaknesses, and mechanisms. Additionally, the paper explores challenging unlearning scenarios, including adversarial prompts, finer-scale concept unlearning, and sequential unlearning, aiming to pave the way for more effective and robust DM unlearning methods. The major contributions include the introduction of a large-scale stylized image dataset, the improvement in the evaluation frameworks for diffusion unlearning, the comprehensive benchmarking over 9 SOTA methods, and the introduction of the three challenging unlearning scenarios, and the novel and previously unknown insights into the different unlearning methods, unlearning settings.

---

> ### Author Rebuttal · Authors · 2024-08-16
>
> # Point-to-Point Response to Reviewer CJtq
>
> We thank the reviewer for the detailed summary of our contributions and the insightful feedback. Below, we present a point-to-point to each question from the reviewer.
>
> **Q1. The necessity of fine-tuning.**
>
> **A1.** As discussed in Section 3, fine-tuning is necessary to equip the model with more clearly defined and controllable knowledge of model generations related to UnlearnCanvas. This can be reflected by the post-generation prediction accuracy 98.8% customized for UnlearnCanvas object/style (see Table A4). Moreover, regarding the computing cost,  fine-tuning of SD 1.5 on UnlearnCanvas required about 500 GPU hours, but fine-tuning is only conducted once. The resulting fine-tuned model is used as an initial state of different unlearning methods.
>
> **Q2. Missing Related Literature**
>
> **A2.** We add the evaluation of the method MACE in the **Table R2** ([link](https://imgur.com/a/okziUxF)) (the last row). As we can see, while MACE generally performs well in achieving high unlearning accuracy and efficiency, it suffers from low retainability and FID scores. We appreciate the reviewer for pointing out this related work, and we will include it in the revised submission.
>
> **Q3: Quantitative Judgment of Stylistic Evaluation**
>
> A3. In response to this concern, we evaluated the stylistic difference between any two styles inside UnlearnCanvas and WikiArt, and the result is plotted in **Figure R2** ([link](https://imgur.com/a/YnYeCM2)). A smaller value inside a cell represents a higher stylistic similarity shared by the corresponding two styles.
> Specifically, we utilized the widely accepted gram matrix-based style loss [1] to evaluate the stylistic differences between two images. This approach involves calculating the "Gram matrices" of the intermediate activations from a pretrained CNN. The Gram matrix is the inner product of the feature maps at a given layer, capturing the statistics of patterns at a particular spatial scale.
> As observed, the low style loss values along the diagonal of UnlearnCanvas indicate high stylistic consistency, while the high off-diagonal values reflect significant inter-style discrepancies. In contrast, WikiArt does not display such patterns, which contributes to the low discrimination between styles as revealed in Table A3 in the supplement of the submission.
>
> > [1] Gatys et al., Image Style Transfer Using Convolutional Neural Networks, CVPR 2016.
>
> **Q4. Implementation Details of the Benchmarked Methods**
>
> **A4.** We apologize for the confusion. The implementation details have been provided in Appendix B, where we believe that all  the information necessary to reproduce the results is available.
>
> **Q5. Question on Figure 6**
>
> **A5.** This is an excellent question. To clarify, in Figure 6, we use the pink bars to represent the unlearning accuracy and the blue bars for unlearning robustness. A larger drop between the pink and blue bars in style unlearning indicates a poorer robustness of style unlearning, rather than the unlearning difficulty. Instead, if we only consider the absolute values of the pink bars, the bars in object unlearning are lower, indicating greater difficulty in unlearning (i.e., lower unlearning accuracy)—consistent with the findings of ESD. However, unlearning accuracy  does not necessarily correlate with robustness. This highlights the need for rigorous evaluation of unlearning robustness, which is an important insight revealed by this experiment.
>
> **Q6. Typos**
>
> **A6.** We appreciate the reviewer for carefully scrutinizing our paper, and we have corrected all the identified typos in the revised version.
>
> **Q7. Possible Solutions or Directions of the Discovered Challenges**
>
> **A7.** This is a thoughtful question, and exploring potential solutions or directions for the challenges revealed in this work is indeed valuable for a benchmark paper. Here are our brief suggestions:
> 1. Retainability challenges, especially CRA: Saliency-based approaches may  offer a solution. Since SalUn performs well in Table 2, a more sophisticated method to identify the most influential weights (in unlearning and retainability-preservation) than SalUn might enhance the  performance.
> 2. Unlearning robustness: Adversarial unlearning is a promising approach, where adversarial attacks are performed on the forget set during unlearning to ensure the process addresses worst-case scenarios. Yet, it may take high computation overhead. Thus, the integration of adversarial unlearning with model saliency (point 2) could be a more effective solution.
> 3. Sequential unlearning: Insights from continual learning could be valuable here. Techniques developed to prevent catastrophic forgetting could provide ideas for preventing the re-occurrence of unlearned knowledge.

---

> > ### Comment · Reviewer_CJtq · 2024-08-20
> > **Update: I raised my score and have some further suggestions.**
> >
> > Thank the authors for the detailed response.
> >
> > I checked the authors' rebuttal, the other reviewers' questions and the general comments. I agree with other reviewers that the experiments on more than one diffusion models (or more stable diffusion variants) are indeed necessary to demonstrate the generalization of this work's findings, even though I did not mention this in my review. At the same time, I believe the authors have made an adequate efforts on solving my questions as well as addressing the generalization issue. I believe this work is a timely efforts on sheding lights on and addressing the critical issues in the field and can provide valuable information to the community. The writing, organization, and plottings are also of the highest quality in my review batch. Therefore, I increased the score and recommend acceptance of this work.
> >
> > In the meantime, this paper can be further improved based on my understanding and here are some further comments.
> >
> > * I recommend the authors to move the sequential unlearning experiments back to the main manuscript, if page limit allows. This part, based on my understanding, are of high significance and the phenomena revealed by it are even more novel than the other two scenarios in this work.
> >
> > * I agree with reviewer CWGN, that a one-or-two-sentence introduction for each benchmarked method is a very good way to depict what this method does, besides Table 1. What I suggested in my original review is trying to include more implementation details for each method in the main manuscript, and thus, I believe these two parts can be joined together and reflected in the revision.
> >
> > * Please try to include my other suggestions, such as the discussion of possible future research directions to solve the challenges in your revision.
> >
> > Thanks.

---

> > > ### Author Response · Authors · 2024-08-20
> > > **Thank You for Raising the Score and Your Acknowledgement on Our Contribution!**
> > >
> > > Dear Reviewer CJtq,
> > >
> > > Thank you very much for your prompt response! We highly appreciate your feedback and your decision to increase the score to 9. Your acknowledgment is encouraging and holds significant value for us. We will include all the discussions we made during the rebuttal into our revised version for sure, including the discussion on the generalization of the results, the details of the benchmarked methods, as well as the possible research directions regarding the challenges revealed by this work. We will also move the sequential unlearning experiment back to the main manuscript if page limit allows. Thank you again for your suggestions!
> > >
> > > Best regards,
> > >
> > > Authors

---

### Official Review · Reviewer_CWGN · 2024-07-25
**This paper presents a dataset and a benchmark for machine unlearning in diffusion models. It addresses a timely and important task and is of high quality overall.**

**Rating:** 6
**Confidence:** 3
**Correctness:** The presented dataset and benchmarkin…
**Clarity:** Yes

**Review:**

### Pros
* Overall, the paper is well-written and easy to follow. The provided dataset addresses several challenges in existing datasets.
* Detailed evaluations and analyses are conducted using nine recently proposed methods on the provided dataset.

### Cons
* As the author mentioned, this paper only tests on a single diffusion model (Stable Diffusion v1.5). It remains unclear whether these results are applicable to other diffusion models. Would there be significant differences in the unlearning results for different diffusion models or various versions of Stable Diffusion?
* The paper evaluates nine MU methods. It would be appreciated if the authors could provide a high-level summary of these methods.
* While the paper is generally easy to follow, some of the concepts are confusing and require further exploration:
    * Line 173, please clarify the definition of "cross-domain." Does this refer to all objects in all styles except for the target style?
    * Line 211, please clarify the setup of unlearning target. Are each of the 20 objects and 60 styles (1200 different unlearning targets in total) individually used as targets for unlearning?
* What prompts are used to evaluate different metrics? Please clarify and discuss the use of prompts in more detail.

**Strengths:**

* Machine unlearning for diffusion models is a timely and crucial task. This paper provides a comprehensive evaluation of recent methods addressing this task and offers intriguing observations based on the evaluation results.
* The authors offer a more comprehensive evaluation of machine unlearning methods, including assessments of both effectiveness and retainability, compared to existing work.

**Additional Feedback:**

No

**Documentation:**

This paper provides sufficient detail on collection, availability and instructions for evaluation pipeline.

**Limitations:**

The authors have adequately addressed the limitations and potential negative societal impact of their work.

**Opportunities For Improvement:**

Refer to review part

**Relation To Prior Work:**

Yes

**Summary And Contributions:**

This paper presents a dataset and a benchmark for machine unlearning in diffusion models. It addresses a timely and important task and is of high quality overall.

---

> ### Author Rebuttal · Authors · 2024-08-16
>
> # Point-to-Point Response to Reviewer CWGN
>
> We thank the reviewer for the detailed feedback. Below, we present a point-to-point reply to each question of the reviewer.
>
> **Q1. This paper only tests on a single diffusion model (SD v1.5). It remains unclear if these results are applicable to other diffusion models. Are there significant differences in the unlearning results for different DMs or various versions of SD?**
>
> **A1.** Thank you for raising this question. To alleviate this concern, we conducted additional experiments on Stable Diffusion (SD) v2.0 and SD v1.4, which are newer and older versions of the SD v1.5 used in our original submission. Please refer to our general response ([**GR**] ([link](https://openreview.net/forum?id=t9aThFL1lE&noteId=IyUyjiXAJh))) for more details and the results are reported in **Table R1** ([link](https://imgur.com/a/qmOe1Dh)).
>
> The additional experimental justification demonstrates the major conclusions in Tab. 2 remain consistent, regardless of the diffusion model type tested. The strengths and weaknesses of each unlearning method, as identified in Tab. 2, also persist across different SD models. Notably, the low IRA (In-Domain Retaining Accuracy) and CRA (Cross-Domain Retaining Accuracy) values in SD v2.0 and SD v1.4 implies the importance of retainability for  a comprehensive evaluation of the performance of the unlearned diffusion models, consistent with our discussion in Lines 263-267. In addition, we also observe that no single method excels across all metrics simultaneously, consistent with our discussion in Lines 279-283. We hope this additional experiment addresses your concerns about the generalizability of our conclusions across different SD model types.
>
> **Q2. Can authors provide a high-level summary of the 9 methods evaluated in this work?**
>
> **A2.** We appreciate this excellent suggestion. Below is a high-level summary of the key unlearning techniques. We will also incorporate this summary in the revised version of our submission.
>
> * **ESD** finetunes DMs by guiding outputs away from a specific concept targeted for erasure, using the idea similar to “classifier-free guidance”.
> * **CA** finetunes DMs by aligning the target concept's distribution with an anchor concept using KL divergence to ablate specific concepts.
> * **UCE** applies closed-form modifications to attention weights. It adjusts linear projection layers for specific text embeddings while preserving other concepts.
> * **FMN** re-steers attention maps to minimize the influence of specific concepts, erasing them while preserving the model's performance on other content.
> * **SalUn** performs unlearning by applying gradient-based weight saliency, focusing on specific model weights rather than the entire model.
> * **SEOT** unlearns concepts by using soft-weighted regularization and inference-time text embedding optimization.
> * **SPM** uses a one-dimensional adapter to erase multiple concepts from DMs.
> * **EDiff** solves a constrained optimization problem to balance retainability and unlearning effectiveness.
> * **SHS** erases data influence by identifying and reinitializing the most sensitive parameters in a model related to the forgetting data, using connection sensitivity analysis.
>
> **Q3. While the paper is generally easy to follow, some concepts require further exploration:**
> * Line 173: "cross-domain." Does this refer to all objects in all styles except for the target style?
> * Line 211: the setup of the unlearning target. Are each of the 20 objects and 60 styles (1200 targets in total) individually used as targets for unlearning?
>
> **A3.** We apologize for any confusion in the original version and provide additional explanations below:
> * **Cross-Domain**: The reviewer's understanding is correct. In our work, the concept domain can be either style domain or object domain. Taking style unlearning for example, the cross-domain retaining accuracy (CRA) is calculated as the average object-wise classification accuracy over the images generated with prompt "A painting of {O} in {S} style", where "O" traverses all the 20 objects and "S" traverses all the styles except for the target style.
> * **The setup of the unlearning target**: In Table 2, either an object or a style is unlearned at a time, resulting in a total of 80 unlearning targets (20 objects + 60 styles). In Table 3, each experiment involves unlearning a combination of an object and a style, leading to 1200 (20 * 60) potential targets. For efficient experimentation, we choose 60 combinations as the unlearning targets in Table  3  to make sure all the styles and objects are covered in the selected subset.
>
> **Q4. What prompts are used to evaluate different metrics?**
>
> **A4.** Throughout our work, we use the prompt template "A painting of {O} in {S} style" to generate images for quantitative evaluation, where "O" can be any of the 20 object options and "S" be any of the 60 styles.
>
> For unlearning accuracy, suppose the unlearning target is "Van Gogh" style, we check the ratio of the images generated with "A painting of {O} in Van Gogh style" classified not into the Van Gogh class, where "O" traverse all the 20 objects. Similarly, when "Dog" is used as an unlearning target, images generated with "A painting of Dog in {S} style", with "S" traversing all 60 styles, are tested to see if they are classified not in the dog class.
>
> To assess in-domain retaining accuracy (IRA) when unlearning a style (e.g., Van Gogh style), we evaluate the ratio of images generated with the template "A painting of {O} in {S} style", with "S" traversing all the styles except the Van Gogh style and "O" traversing all objects, which are correctly classified into the corresponding style class. The similar idea applies to the IRA evaluation of object unlearning.
>
> For evaluating cross-domain retaining accuracy (CRA), we use a similar approach to IRA but the evaluation focuses on the concept domain different from the unlearning target, as detailed in A3 above.

---

> ### Author Response · Authors · 2024-08-28
> **Thank you and genuine request for follow-up discussions!**
>
> Dear Reviewer CWGN,
>
> We extend our heartfelt appreciation for your dedicated review of our paper. As outlined in the recent NeurIPS email communication, we are fully aware of the commitment and time your review entails. Your efforts are deeply valued by us.
> During the rebuttal, we have made substantial effort in addressing all the reviewers’ questions, e.g., the evaluation using another diffusion models, the variety of the styles, adding new methods to the benchmark, the quantitative comparison between UnlearnCanvas and WikiArt; see a summary of extended experimental justifications/results in [general response](https://openreview.net/forum?id=t9aThFL1lE&noteId=IyUyjiXAJh).
>
> As the author-reviewer discussion phase draws to an end soon, we wish to extend a respectful request for your feedback about our general response and individual responses. Your insights are of immense importance to us, and we eagerly anticipate your updated evaluation. Should you find our responses informative and useful, we would be grateful for your acknowledgment. Furthermore, if you have any further inquiries or require additional clarifications, please don't hesitate to reach out. We are fully committed to providing additional responses during this crucial discussion phase.
>
> We sincerely thank you for your continued support and consideration. Your expertise is pivotal to the advancement of our research.
>
> Best regards,
>
> Authors

---

### Author Rebuttal · Authors · 2024-08-16

# Highlighted General Response

We extend our gratitude to all the reviewers for their invaluable comments and constructive suggestions. Below, we present a general response (GR) related to the additional experiments we conducted in the rebuttal.

## GR - Additional Experiment Results with UnlearnCanvas on More Diffusion Models.

Some reviewers expressed concerns that the experiments in this work were conducted with only a single diffusion model, raising questions about the generalizability of our conclusions. We acknowledge these concerns and have therefore conducted additional experiments to provide further validation.

Specifically, we re-ran the experiments related to the main results reported in Table 2 on the new Stable Diffusion (SD) model versions, SD v1.4 and SD v2.0 (a prior and subsequent version of the SD v1.5 model used in our study). Due to the time constraint during the rebuttal phase, we selected 20 out of the 60 styles and 10 out of the 20 objects proposed in UnlearnCanvas for these experiments. Our goal is to demonstrate the validity and reliability  of our findings presented in the submission. The latest results are shown in **Table R1** ([link](https://imgur.com/a/qmOe1Dh)).

Several key observations can be drawn  from this new comparison: (1) The primary conclusions presented in the original Table 2 (Lines 263-283) remain valid. Retainability continues to be crucial for a comprehensive assessment for unlearning in diffusion models, as we observed even worse retainability with the newer model (SD v2.0). Moreover, CRA (Cross-Domain Retaining Accuracy) remains generally more challenging to maintain than IRA (In-Domain Retaining Accuracy). In the results for both SD v1.4 and SD v2.0, no single method consistently outperforms others across all metrics, consistent with our finding in Lines 279-283. (2) Changing the base model can result in performance fluctuations for each unlearning method, yet the  relative performance comparison  of these  methods are consistent. Specifically, we note that ESD remains the most well-rounded method for style unlearning, while SalUn performs better in object unlearning. The unlearning effectiveness of the method SEOT still requires improvement, and the CRA of SHS is notably lower compared to other methods.

We hope these additional experiments can address the reviewers' concerns, and we will enrich the above experimental study as well as its discussion in our revision.

## Summary of Additional Experiments

Dear reviewers,

We are glad to receive your valuable and constructive comments. We have made a substantial effort to clarify your doubts and enrich our experiments in the rebuttal phase. Below, we provide a summary of the additional results we have conducted. **All the new results are compiled in the one-page PDF, with links to each table/figure provided after its reference for your convenience.**

* [**Reviewer CWGN**](https://openreview.net/forum?id=t9aThFL1lE&noteId=MgFAZC0te9):
    * We conducted additional experiments on evaluating more Stable Diffusion variations (SD v2.0 & SD v1.4) with UnlearnCanvas following the setting of Table 2 in our work and demonstrated that the conclusions from Table 2 can be generalized to other stable diffusion models than SD v1.5. The results are shown in **Table R1** ([link](https://imgur.com/a/qmOe1Dh)).

* [**Reviewer Cjtq**](https://openreview.net/forum?id=t9aThFL1lE&noteId=Xn5jDmieSk):
    * We have enriched the method evaluated in this work and added thea new method MACE. The results are shown in **Table R2** ([link](https://imgur.com/a/okziUxF)).
    * We conducted a quantitative comparison on the intra-style stylistic consistency and inter-style discrepancy between UnlearnCanvas and WikiArt. The results are plotted in **Figure R2** ([link](https://imgur.com/a/YnYeCM2)).

* [**Reviewer MenV**](https://openreview.net/forum?id=t9aThFL1lE&noteId=1oSzJ4RXaL):
    * We evaluated more Stable Diffusion variations (SD v2.0 & SD v1.4) with UnlearnCanvas following the setting of Table 2 (the main results) in our work and demonstrated that the conclusions can be generalized to other stable diffusion models than SD v1.5. The results are shown in **Table R1** ([link](https://imgur.com/a/qmOe1Dh)).
    * We re-evaluated the answer set associated with Table 2 with a different style/classifier, i.e., a ResNet-101 finetuned on UnlearnCanvas, to demonstrate that the results will not be affected by the choice of the type of the classifier. The results are shown in **Table R3** ([link](https://imgur.com/a/WIHRpoc)).

* [**Reviewer T5uq**](https://openreview.net/forum?id=t9aThFL1lE&noteId=lb9U6rf8t6):
    * We demonstrate that images of different types within a single artistic style (Van Gogh style in our demonstration) can be generated via Fotor, the tool we used to curate UnlearnCanvas. The results are shown in **Figure R1** ([link](https://imgur.com/a/57UvoQO)).
    * We provide a single-score evaluation and the relavtive ranking for all the methods based on the rankings revealed in Figure 1(b) to ease the direct comparison. The results are reported in **Table R4** ([link](https://imgur.com/a/QUbqWB2)).

---

### Decision · Program_Chairs · 2024-09-26

**Decision:**

Accept (Poster)

**Comment:**

This paper introduces the UnlearnCanvas dataset, designed to evaluate the machine unlearning (MU) capability of diffusion models (DM). UnlearnCanvas considers both style and object unlearning scenarios and provides a dataset with high-resolution images and detailed class labels. The work also presents eleven new benchmarks to evaluate nine machine unlearning methods.

Reviewers initially raised questions about the methodology details and requested clarifications on various aspects of the study. During the rebuttal period, the authors effectively addressed the reviewers' concerns. The authors provided clarifications and additional information that satisfied the reviewers' questions.

Given the authors' successful addressing of the concerns during the rebuttal period, and considering the potential contribution of this research to the field of machine unlearning, the consensus is that this work will be beneficial to the machine unlearning field and the Area Chair has decided to accept this paper for the conference.